# Vitamin D Status and Incidence of SARS-CoV-2 Reinfections in the Borriana COVID-19 Cohort: A Population-Based Prospective Cohort Study

**DOI:** 10.3390/tropicalmed10040098

**Published:** 2025-04-06

**Authors:** Salvador Domènech-Montoliu, Laura López-Diago, Isabel Aleixandre-Gorriz, Óscar Pérez-Olaso, Diego Sala-Trull, Alba Del Rio-González, Maria Rosario Pac-Sa, Manuel Sánchez-Urbano, Paloma Satorres-Martinez, Juan Casanova-Suarez, Cristina Notari-Rodriguez, Raquel Ruiz-Puig, Gema Badenes-Marques, Laura Aparisi-Esteve, Carmen Domènech-León, Maria Angeles Romeu-Garcia, Alberto Arnedo-Pena

**Affiliations:** 1Medical Direction University Hospital de la Plana, 12540 Vila-Real, Spain; pttcarmen@hotmail.com; 2Clinical Analysis Service University Hospital de la Plana, 12540 Vila-Real, Spain; lopez_laudia@gva.es (L.L.-D.); aleixandre_isagor@gva.es (I.A.-G.); 3Microbiology Service University Hospital de la Plana, 12540 Vila-Real, Spain; perez_oscola@gva.es; 4Emergency Service University Hospital de la Plana, 12540 Vila-Real, Spain; saladiego2@gmail.com (D.S.-T.); manu.msu@gmail.com (M.S.-U.); palomasatmar@gmail.com (P.S.-M.); notari_cri@gva.es (C.N.-R.); raquelruizpuig@gmail.com (R.R.-P.); gemabamar@gmail.com (G.B.-M.); 5Health Centers I and II, 12530 Borriana, Spain; delrio_alb@gva.es; 6Public Health Center, 12003 Castelló de la Plana, Spain; charopac@gmail.com (M.R.P.-S.); aromeu96@gmail.com (M.A.R.-G.); 7Nursing Service University Hospital de la Plana, 12540 Vila-Real, Spain; juancasanova83@gmail.com; 8Carinyena Health Center, 12540 Vila-Real, Spain; lauraaparisiesteve@gmail.com; 9Department Medicine, Universidad CEU Cardenal Herrera, 12006 Castelló de la Plana, Spain; carmendomenech04@gmail.com; 10Epidemiology and Public Health (CIBERESP), 28029 Madrid, Spain; 11Department of Health Science, Public University Navarra, 31006 Pamplona, Spain

**Keywords:** COVID-19, SARS-CoV-2 infection, vitamin D, reinfection, cohort, prospective, population-based

## Abstract

A deficient vitamin D (VitD) status has been associated with SARS-CoV-2 infections, severity, and mortality. However, this status related to SARS-CoV-2 reinfections has been studied little. Our aim was to quantify the risk of reinfections considering VitD status before reinfection. Methods: We performed a population-based prospective cohort study in Borriana (Valencia Community, Spain) during 2020–2023, measuring 25-hydroxyvitamin D [25(OH)D] levels by electrochemiluminescence. Cox proportional hazards models were employed. Results: Of a total of 644 SARS-CoV-2 cases with confirmed laboratory tests, 378 (58.9%) were included in our study, with an average age of 38.8 years; 241 were females (63.8%), and 127 reinfections occurred (33.6%). SARS-CoV-2 reinfection incidence rates per 1000 person-days by VitD status were 0.50 for a deficient status (<20 ng/mL), 0.50 for an insufficient status (20–29 ng/mL), and 0.37 for a sufficient status (≥30 ng/mL). Compared with a sufficient VitD status, adjusted hazard ratios were 1.79 (95% confidence interval [CI] 0.89–3.59) for a deficient status and 1.59 (95% CI 1.06–2.38) for an insufficient status with a significant inverse dose–response (*p* = 0.02). These results can help improve nutritional actions against SARS-CoV-2 reinfections. Conclusions: These results suggest that a VitD status lower than 30 ng/mL showed a higher risk of SARS-CoV-2 reinfection. Achieving and maintaining a sufficient VitD status is recommended to prevent reinfections.

## 1. Introduction

After the devastating COVID-19 pandemic, there was a rising interest in vitamin D (VitD)’s role due to its antimicrobial properties, its immunomodulatory action [1,2,3], and its effect on respiratory infections, including tuberculosis [4,5]. In fact, using the electronic database PubMed, more than 2000 references have reported on this VitD-COVID-19 relationship from 2020 to 2025.

Different observational epidemiologic studies between VitD status and SARS-CoV-2 infections have been performed, regarding prevention, morbidity, and mortality [6,7,8,9,10,11,12]. In 2022, Albergamo and co-authors [13] summarized the situation, considering that a deficient VitD status can be associated with COVID-19 severity, hospitalization, and death. However, deeming VitD deficiency a risk factor for COVID-19 incidence is controversial [14]. In addition, there are contradictory results of several clinical trials about VitD supplementation to prevent SARS-CoV-2 infections [15].

Regardless the heterogeneity of the studies, with different populations and protocols [16], the research of VitD and SARS-CoV-2 infections seems complex, including the reverse causality, (what comes first, VitD deficiency or the infection?), different techniques for measuring VitD levels, conjugate VitD measurement [17], VitD receptor genetic polymorphisms [18], VitD-binding proteins [19], personal VitD response index [20], inadequate levels of VitD status, infection itself as risk factor for decreased VitD status [21], seasonal changes in VitD status, and a considerable number of other potential confounding factors, such as age, sex, ethnicity, body mass index, lifestyle, chronic diseases, and SARS-CoV-2 exposures. Regarding SARS-CoV-2 reinfections, different definitions have been used; subclinical or asymptomatic infections can take place, SARS-CoV-2 variants present reinfection differences, and a sufficient and adequate follow-up of patients is needed to detect reinfections [22,23,24].

The relationship between VitD status and SARS-CoV-2 reinfections has been less studied. In general, it has been found that there are low rates of SARS-CoV-2 reinfections, except for the Omicron variants, with lower severity than first infections, possible under-diagnosis, and a deficiency of VitD not considered [25,26,27].

Our hypothesis is that in patients who had suffered a first SARS-CoV-2 infection, a lower than 30 ng/mL 25-hydroxyvitamin D (25(OH)D) level in serum measured after the infection could have been a risk factor for a SARS-CoV-2 reinfection. The aim of our research is to estimate the relationship between serum VitD status and the incidence of reinfection in the Borriana COVID-19 cohort.

## 2. Materials and Methods

### 2.1. Description

We performed a population-based prospective cohort study extracted from the Borriana COVID-19 cohort. The study began with the COVID-19 outbreak of the Fallas Festival in March–June 2020 in the city of Borriana (Valencia Community, Spain) [28]. In October 2020, a cross-transversal study was performed, including participants with positive laboratory tests for SARS-CoV-2, and serum levels of [25(OH)D] were measured [29]. In June 2022, a new survey was implemented with all the participants who had completed laboratory tests for COVID-19 surveyed [30], and 25(OH)D levels were newly measured. In December 2022, a sample of the participants in the June 2022 survey was chosen to study cellular immunity [31]. In addition, a follow-up of the participants using the register of primary healthcare was carried out by health staff of the University Hospital de la Plana, and the computer application outpatient care of the Valencia Health Agency (ABUCASIS) was queried from January 2020 to August 2023 with the aim to detect new COVID-19 cases, reinfections, sequelae, long COVID-19 patients, and deaths.

Figure 1 shows the flow chart of the study presented from the first survey in 2020. In the June 2022 survey, 722 participants were included, with 386 cases and 335 no-cases. The follow-up of this cohort from January 2020 to August 2023 found 644 SARS-CoV-2 cases confirmed by the laboratory, with 75 no-cases and 3 suspected SARS-CoV-2 cases. From these cases, there were 156 SARS-CoV-2 reinfections, 456 no-reinfections, and 35 suspected reinfections. In our study, 378 participants—all SARS-CoV-2 confirmed cases—were included, with 127 reinfections and 251 no-reinfections.

Serum VitD status was measured two times in October 2020 and June 2022. The mean of VitD in the first survey was 29.6 ± 9.8 ng/mL, and in the second, this was 30.3 ± 9.4 ng/mL with no significant differences (*p* = 0.508). All the participants had their VitD status measured after the first SARS-CoV-2 infection. For participants without SARS-CoV-2 reinfection, their reported VitD status was the closest to the finish of the follow-up. For participants with SARS-CoV-2 reinfections, their reported VitD status was the closest before the first reinfection and at least three weeks before this reinfection. The time between VitD status determination and SARS-CoV-2 reinfection or finishing the follow-up had a mean of 284 ± 144.2 days. Levels of 25(OH)D were measured by electrochemiluminescence-based assay Elecsys of Roche Diagnostic [32]. VitD status was defined as severely deficient (0–9 ng/mL), deficient (10–19 ng/mL), insufficient (20–29 ng/mL), or sufficient (≥30 ng/mL) [33]. These analyses were performed at the Clinical Laboratory Service of La Plana University Hospital, Vila-real (Spain).

SARS-CoV-2 reinfection was defined as a new SARS-CoV-2 infection more than 60 days after the previous SARS-CoV-2 infection and confirmed by polymerase chain reaction test (PCR of different platforms and commercial kits) or a rapid antigen test (RAT) [34]. The first infection must have been confirmed by PCR, RAT, or positive anti-nucleocapsid IgG determinations.

Inclusion criteria included reinfection SARS-CoV-2 cases with a laboratory confirmation test and VitD status measured at least 3 weeks before the first reinfection. Only the first reinfection was included in the analysis. No-reinfection cases included participants whose VitD status was measured and in whom no reinfection occurred during the study period.

Exclusion criteria included SARS-CoV-2 cases with VitD status measured before the first infection, a reinfection not confirmed by laboratory test, a VitD status measured less than 3 weeks before reinfection, reinfections occurring less than 60 days after the first infection, and loss of follow-up after VitD measurement.

Laboratory confirmatory tests from different surveys included the following determinations: anti–SARS-CoV-2 spike IgG antibodies and IgG and IgM anti-nucleocapsid antibodies were measured by chemiluminescence microparticle immunoassay (CMIA AlinityI serie, Abbot, Chicago, IL, USA) [35] in the third survey and antibodies against SARS-CoV-2 nucleocapsid protein N were studied by electrochemiluminescence immunoassay [36] in the first and second surveys. For the detection of the SARS-CoV-2 virus, reverse transcription polymerase chain reaction (RT-PCR) and rapid antigen tests (RATs) from different trademarks and platforms were employed. The Microbiology Service Laboratory of La Plana University Hospital (Vila-real, Spain) performed RT-PCR molecular-based tests laboratory tests (Genexpert, Roche Diagnostics, Simplexa, Barcelona, Spain), and RATs were performed at home by the participants.

The surveys consisted of a questionnaire for all participants to gather socio-demographic characteristics, health status, SARS-CoV-2 infections and reinfections, medical attention, lifestyle, and risk factors for SARS-CoV-2 infection, such as body mass index (kg/m^2^), previous chronic diseases, and COVID-19 exposures. Telephone and face-to-face interviews were used to fill in questionnaires and were carried out by health staff of the Health Centers of Borriana and Vila-real, the Emergency Service of University Hospital de la Plana, and the Public Health Center of Castellon.

### 2.2. Statistical Methods

In the descriptive analysis, mean, standard deviation, and ranges were employed with Chi^2^ and Fisher exact tests for comparison of qualitative variables and Kruskal–Wallis tests for quantitative variables. A test of trend was used to study the dose–response of 25(OH)D levels and SARS-CoV-2 reinfections. The time-to-event (reinfections) was estimated in person-days, considering the days from the first SARS-CoV-2 infection to the first SARS-CoV-2 reinfection for the reinfection cases, and the days from the first infection to the finish of the follow-up for no-reinfections cases. We calculated the rate of SARS-CoV-2 reinfection by dividing the number of reinfection cases by the person-days observed in the follow-up, considering the 25(OH)D levels. Multivariable Cox proportional hazard models were used, considering SARS-CoV-2 reinfection as the dependent variable and VitD status as the predictor variable. SARS-CoV-2 reinfection incidence per 1000 person-days regarding VitD levels was estimated. Crude and adjusted hazard ratios (HRs) and 95% confidence intervals (CIs) were estimated. The directed acyclic graphs (DAGs) method was used for the control of potential confounding factors with the DAGitty^®^ program (version 3.1) [37]. After a review of the COVID-19 literature, potential confounders included age, sex, obesity, alcohol consumption, smoking habit, number of doses of SARS-CoV-2 vaccine, exposures to COVID-19, and chronic diseases (Figure 2). The Stata^®^ program version 14.2 was used for all statistical calculations.

### 2.3. Sensitivity Analysis

As an alternative to multivariable Cox proportional hazard models, we used another statistical approach, and we employed inverse probability weighted regression [38]. This approach permits estimations of crude and adjusted cumulative incidence of SARS-CoV-2 reinfections and relative risks (RRs), considering the number of SARS-CoV-2 reinfections divided by the total number of participants as exposed.

This study was approved by the Ethics Committee of University Hospital la Plana (registry number 2961). All participants or the parents of minors provided informed written consent to be included in the study.

## 3. Results

Of 644 SARS-CoV-2 cases in the Borriana cohort during the period of study, 153 SARS-CoV-2 reinfections were reported, as well as 35 suspected reinfections and 456 participants without reinfections. When considering inclusion and exclusion criteria, we found 127 reinfections and 251 people without reinfection; a total of 378 participants were included in the study (58.7%). The average age of participants was 38.8 ± 16.6 years (range 1–75 years), with 241 females (63.8%) and 137 males (36.2%). A total of 127 cases of SARS-CoV-2 reinfections occurred with a cumulative incidence rate of 33.6% (127/378). Considering the total number of participants, the cumulative rate of infections rose 89.2% (644/722).

Characteristics of SARS-CoV-2 reinfections and no-reinfection participants are shown in Table 1. Reinfection-case patients were younger than in no-reinfection cases, with higher rates of reinfection in the 1–24 years age group (37.4%) compared with the other age groups, but without significant differences. No differences were found between sexes, presence of chronic diseases, prevalence of obesity, or alcohol consumption. Considering specific chronic diseases such as cardiovascular disease, arterial hypertension, diabetes mellitus, hypothyroidism, asthma, and allergic rhinitis, they did not show significant differences between the two groups. More doses of the SARS-CoV-2 vaccine were administered for the no-reinfection group, and non-smokers were more frequent in this group. COVID-19 exposures did not present differences regarding reinfections.

Regarding SARS-CoV-2 reinfection time distribution, the first reinfections occurred in January and March 2021 (4 cases), with few reinfections until December 2021 (14 cases), and an increase from January to March 2022 (59 cases). From March to August 2022, reinfections remained high (36 cases), with a slow decrease until December 2022 (9 cases). From January to August 2023, reinfections decreased (five cases). In a reinfection case that occurred in March 2022, the Omicron variant BA.2.9 was isolated.

With respect to the clinical course, the first SARS-CoV-2 infections were symptomatic in the majority of participants, with 89.9% (340/378), reaching 88.8% in the no-reinfections group and 92.1% in the reinfections group (*p* = 0.37). In general, disease episodes were mild. The hospitalization rate was 4.8% (18/378)—4.4% in the no-reinfections group and 5.5% in the reinfections group (*p* = 0.62). The course of the disease was towards recovery, although the incidence of sequelae, such as long COVID syndrome, was high, with 28.0% (106/378) of all participants, 27.5% in the no-reinfections group, and 29.1% in the reinfections group (*p* = 0.80). The age group of participants 60 years and older was the most affected, with 90.2% of symptomatic infections, 14.6% with hospitalizations, and 34.1% developing sequelae.

Among the participants, the prevalence of VitD status was severely deficient in 0.3% (1/378), deficient in 7.1% (27/378), insufficient in 47.9% (181/378), and sufficient in 44.7% (169/378). Severely deficient and deficient levels were put together considering their small number (Table 2). The mean of 25(OH) D was higher in the no-reinfection group, at 30.4 ng/mL vs. 29.0 ng/mL (*p* = 0.17) in the reinfection group, but this is not significant. Considering 25(OH)D levels, the reinfections were more frequent in lower than high levels, and comparing sufficient with deficient and insufficient levels, a significant difference was found (*p* = 0.04).

Considering the person-days of participants from the first infections (Table 3), a crude analysis of Cox proportional hazard models found a higher SARS-CoV-2 reinfection incidence rate per 1000 person-days for deficient and insufficient VitD statuses, at 0.50 vs. 0.37 for sufficient VitD status. In total, the reinfection incidence rate was 0.44 per 1000 person-days.

The risk of SARS-CoV-2 reinfections considering VitD status is shown in Table 4. Crude HRs of reinfections were higher for the low levels of 25(OH)D, but no significant differences were observed. However, an adjusted HR presented a significant inverse dose–response (*p* = 0.02) with HR = 1.79 (95% CI 0.89–3.59) level <20 ng/mL, and HR = 1.59 (95% CI 1.06–2.38) level 20–29 ng/mL when compared with ≥30 ng/mL level. Deficiency and insufficiency in VitD status had a 61% significantly higher risk of reinfections compared with sufficiency status (HR = 1.61, 95% CI 1.09–2.39).

In the sensitivity analysis, inverse probability weighted regression analysis is used (Table 5). In the crude analysis, reinfection incidence rates were lower for sufficient VitD status, at 0.28 vs. 0.39 and 0.38 for deficient and insufficient VitD statuses. In the adjusted analysis, the reinfection incidence rate was lower for the sufficient status, at 0.25 vs. 0.40. Tests for the trend of 25(OH)D levels and reinfections were significant in the crude (*p* = 0.04) and adjusted analysis (*p* = 0.01), indicating a dose–response effect, lower 25(OH)D levels, and an increase in reinfections.

Crude and adjusted relative risks (RRs) of reinfections are shown in Table 6. In the crude analysis, a deficient VitD status had a greater risk than insufficient and sufficient levels. In the adjusted analysis, an insufficient status presented more risk of reinfections compared with a sufficient status (RR = 1.57, 95% CI 1.15–2.16). Comparing sufficient VitD status with deficient and insufficient levels, the RR was 1.39 (95% CI 1.02–1.85) in the crude analysis, and the RR was 1.57 (95% CI 1.17–2.15) in the adjusted analysis. Low 25(OH)D levels presented a 57% greater risk of reinfections than sufficient levels.

## 4. Discussion

Our results suggested that VitD status was associated with SARS-CoV-2 reinfection: having levels lower than 30 ng/mL of 25(OH)D levels increased the risk of reinfection. An inverse dose–response relationship between 25(OH)D levels and risk of reinfection was observed in both Cox proportional hazards models and sensitivity analysis.

In our participants, VitD status was higher than in other studies about VitD in Spanish and worldwide populations [39,40,41]. This may presume a lower incidence of reinfection. On the other hand, the cumulative incidence of reinfection was, to some extent, reduced, considering that the total number of infections was very high. In fact, the incidence rate of reinfections was lower for the study period from October 2020 to August 2023, at 0.44 per 1000 person-days. In studies of SARS-CoV-2 reinfections, the range of cumulative incidence rates fluctuated between 0.20 and 0.66 per 1000 person-days [42,43,44,45], but they lacked prolonged follow-up. In addition, our participants were widely vaccinated with three or more doses against SARS-CoV-2. Then, the so-called hybrid immunity (booster vaccination plus natural infection) might have decreased the risk of reinfection [46]. However, most of the reinfections occurred during the period when the Omicron variant was predominantly present [24,47]. In our study, only one case of the Omicron variant was isolated, but during the period with the highest cases of reinfections—that is, December 2021 to August 2022—the Omicron variant was the most predominant in Spain [48].

Our results are in line with the study by Chen and co-authors [49] regarding lower recurrence of SARS-CoV-2 infections in an elderly population with 25(OH)D ≥ 30 ng/mL. Abu Fanne and co-authors found in an observational retrospective study [50] that patients with SARS-CoV-2 reinfection have lower 25(OH)D levels than patients without reinfections.

Regarding the risk of SARS-CoV-2 infection and VitD status, in cohort studies with VitD status measured before the infection, Oristrell and co-authors found that participants with 25(OH)D levels of 30 ng/mL and above before SARS-CoV-2 infection had a lower risk of SARS-CoV-2 infection [51]. Other cohort studies with VitD status measured before the SARS-CoV-2 infection have found that a concentration of 25(OH)D below the level of 30 ng/mL increased the risk of SARS-CoV-2 infection [52,53], but in another cohort study, the higher risk of infection was associated with levels below 12 ng/mL [54]. In a case–control study in Israel, low VitD levels were associated with a higher risk of SARS-CoV-2 infections [55]. In addition, in an observational study in the United States, Kaufman and co-authors [56] demonstrated a strong inverse association between circulating VitD levels and SARS-CoV-2 positivity. In Egypt, a case–control study of children and adolescents found that VitD deficiency was associated with an increased risk for COVID-19 infection, and the FokI FF genotype was more frequent in cases than controls [57]. On the other hand, severity and mortality from COVID-19 were lower in those patients with a 25(OH)D level of 30 ng/mL and above, tested before infection [58,59], and high doses of VitD in COVID-19 patients had a reduction in mortality [60,61]. In the United States, the Nurses’ Health Study II found an association between higher predicted circulating 25(OH)D concentrations and a lower risk of SARS-CoV-2 infection [62].

However, other observational studies found that VitD status was not associated with SARS-CoV-2 infections—for instance, in a cohort of healthcare workers in New York [63], in the California general population [64], and in a cohort of patients in Italy [65]. Crandell and co-authors found a minimal protection of VitD status on COVID-19 test positivity in a study of electronic health records [66]. In addition, COVID-19 severity was not associated with VitD status in Mongolia [67]. The evidence of the effective VitD treatment in COVID-19 has been considered low [68], and a Mendelian randomization study did not find an association between 25OHD levels and COVID-19 susceptibility, severity, or hospitalization [69].

Concerning VitD supplementation to prevent SARS-CoV-2 infection, several clinical trials have shown contradictory results. In one study [70], VitD decreased SARS-CoV-2 infections in healthcare workers in Mexico. However, according to three studies, VitD supplementation in the general population [71,72] or healthcare workers [73] was not associated with a reduction in the risk of SARS-CoV-2 infection. On the other hand, some limitations to VitD supplements have been indicated according to a few clinical trials, recommending an individual approach to participants and based on 25(OH)D concentrations [74,75]. However, a recent meta-analysis suggests a protective role of VitD on COVID-19 incidence, ICU admission [12], and severity [76], although controversies continue about the impact of VitD on COVID-19 disease [16,77,78,79].

The effects of VitD on COVID-19 are based on its action on innate and adaptive immunity through several mechanisms that can protect against bacterial and viral infections. These mechanisms include the modulation of dendritic cells, macrophages, T cells, and B cells; anti-inflammatory function by decreasing pro-inflammatory cytokines and interleukins and increasing anti-inflammatory ones; antimicrobial actions by cathelicidin and β-defensin 4A; complement activation, and the induction of autophagy [2,80,81,82,83]. Furthermore, these effects of VitD have been supported by studies in animal models [84,85,86].

These are the strong points of our study: Firstly, we presented a prospective cohort design with a measure of VitD status before reinfection. Secondly, VitD status was measured at the same time for all participants. Thirdly, the participants in this population-based design were representative of the general population. Fourthly, control of confounding factors and hazard ratios was performed. Finally, the results of the sensitivity analysis were aligned with the first results.

This study has some limitations, though. First, undetected SARS-CoV-2 reinfections may occur with a potential misclassification bias. Second, the time between VitD determination and SARS-CoV-2 reinfection may change VitD status. Third, we do not have information about the genetic polymorphisms of vitamin D in our participants [87]. Fourth, the participation rate decreased compared to previous surveys. Fifth, regarding the generalization of the results, participants suffered a mild COVID-19 illness, although the incidence of sequelae was elevated [88]. Sixth, residual confounding could be present. Seventh, no reinfections of SARS-CoV-2 variants were obtained; however, the Omicron variant was the most prevalent during the study period [89]. Eighth, the technique used for determining vitamin D level tends to underestimate the total 25-hydroxyvitamin D [25(OH)D] levels [90]. In this context, the relationship between vitamin D status and SARS-CoV-2 reinfections could be decreased. The direction of this information bias would be towards the null hypothesis. Compared with the liquid chromatography–tandem mass spectrometric (LC–MS/MS), the gold standard for measuring vitamin D status [91], the technique employed in this study has a lower accuracy. However, it is considered acceptable for use in clinical laboratories, with an unlikely risk of misclassifying vitamin D status [92]. Finally, COVID-19 is a new disease, and some circumstances may not have been taken into account in this study.

Our results suggest that having an insufficient or deficient VitD status increases the risk of SARS-CoV-2 reinfection. However, the hazard risks were moderate [93], suggesting that other factors could play a role in the incidence of SARS-CoV-2 reinfection, including VitD conjugate, VitD polymorphism, or substances with complementary actions on VitD effects, such as magnesium and zinc [94,95,96,97].

The results of this study can help improve nutritional actions against SARS-CoV-2 reinfections by achieving and maintaining sufficient VitD status. Continuation with research in this area is guaranteed in order to prepare for potential future epidemics. In this respect, considering the difficulty of implementation and contradictory results of clinical trials, an observational approach, such as a prospective cohort study, may be adequate to obtain more conclusive responses to VitD’s effect on the population’s health.

## 5. Conclusions

Insufficient and deficient VitD statuses were both associated with a higher risk of SARS-CoV-2 reinfection with a dose–response relationship, consistent with some prospective studies on VitD and SARS-CoV-2 infections. Achieving and maintaining a sufficient VitD status is recommended.

## Figures and Tables

**Figure 1 tropicalmed-10-00098-f001:**
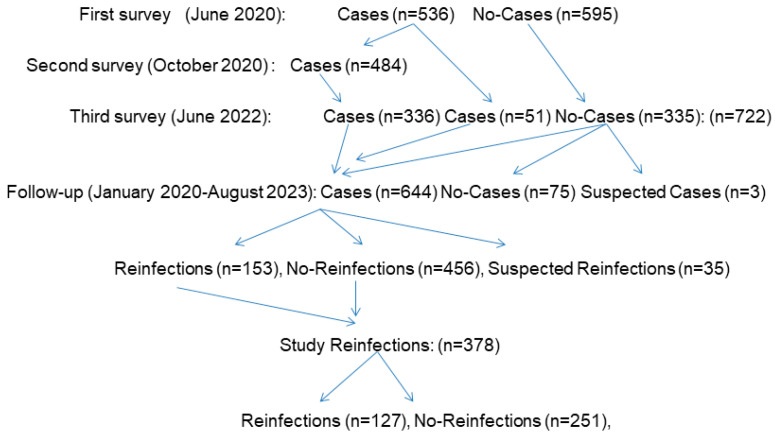
Flow chart for the vitamin D status and SARS-CoV-2 reinfections.

**Figure 2 tropicalmed-10-00098-f002:**
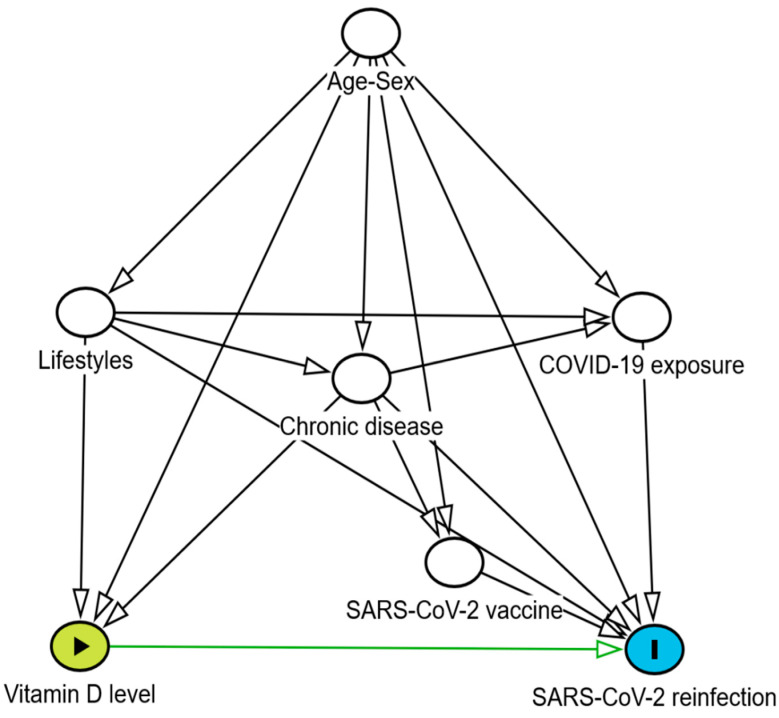
A directed acyclic graph diagram: vitamin D level (exposure) and SARS-CoV-2 reinfection (outcome), adjusted for age–sex, lifestyles, chronic disease, COVID-19 exposure, and SARS-CoV-2 vaccination (confounding factors).

**Table 1 tropicalmed-10-00098-t001:** Characteristics of the participants considering SARS-CoV-2 reinfections and no-reinfections.

Variables	Reinfections *n* = 127 N (%)	No-Reinfections *n* = 251 N (%)	*p*-Value
**Age (years) + SD ^1^**	37.9 ± 17.0	39.2 ± 16.4	0.53
**Age 1–24 years**	34 (26.8)	57 (22.7)	0.70
**25–44 years**	43 (33.9)	90 (35.9)	
**45 years and over**	50 (39.7)	104 (41.4)	
**Sex Female**	82(64.6)	159(63.3)	0.91
**Male**	45(35.4)	92 (36.7)	
**Chronic disease ^2^ Yes**	48 (38.4)	95 (38.2)	1.00
**Chronic disease No**	77 (61.6)	154(61.8)	
**Cardiovascular disease Yes**	14 (11.3)	37 (14.9)	0.42
**No**	110 (88.7)	212 (85.1)	
**Arterial hypertension Yes**	12 (9.8)	24 (9.6)	1.00
**No**	111 (90.2)	225 (90.4)	
**Diabetes mellitus Yes**	3 (2.4)	6 (2.4)	1.00
**No**	121 (97.6)	243 (97.6)	
**Hypothyroidism Yes**	7 (5.7)	12 (4.8)	0.80
**No**	117 (94.3)	237(95.2)	
**Ashma Yes**	3 (2.4)	14 (5.6)	0.20
**No**	121 (97.6)	235 (94.4)	
**Allergic rhinitis Yes**	9 (7.3)	10 (4.0)	0.21
**No**	115 (92.7)	239 (96.0)	
**Obesity ^3^ BMI ^4^ ≥ 30**	24 (18.9)	51 (20.7)	0.79
**BMI < 30**	103 (81.1)	195(79.3)	
**Alcohol consumption ^5^ Yes**	28 (22.8)	50 (20.7)	0.59
**Alcohol consumption No**	95 (77.2)	196 (79.3)	
**Never smoked ^6^**	70 (42.6)	155 (63.3)	0.31
**Current smoker and ex-smoker**	52 (57.4)	90 (36.7)	
**Doses SARS-CoV-2 vaccine**			0.10
**0**	6 (4.7)	5 (2.0)	
**1**	11 (8.7)	11 (4.4)	
**2**	40 (31.5)	70 (27.9)	
**3–4**	70 (55.1)	165 (65.7)	
**Family COVID-19 case Yes**	83(65.4)	151 (60.2)	0.37
**Family COVID-19 case No**	44 (34.5)	100 (39.8)	
**High exposure COVID-19 case ^7^ Yes**	70 (55.6)	199 (60.3)	0.44
**High exposure COVID-19 case No**	56 (44.4)	98 (39.7)	

^1^ SD = standard deviation. ^2^ Chronic disease—missing information for 4 participants. ^3^ Obesity—missing information for 5 participants. ^4^ BMI = body mass index. ^5^ Alcohol consumption—missing information for 9 participants. ^6^ Smoking habit—missing information for 11 participants. ^7^ High exposure COVID-19 case—missing information for 5 participants.

**Table 2 tropicalmed-10-00098-t002:** Distribution of VitD status in SARS-CoV-2 reinfection cases and no-reinfection cases.

Vitamin D	Reinfections	No-Reinfections	Total	
**Three levels**	N = 127 (%)	N = 251 (%)	N (%)	*p*-value
**0–19 ng/mL**	11 (8.7)	17 (6.8)	28 (39.2)	0.10
**20–29 ng/mL**	69 (54.3)	112 (44.6)	181 (38.1)	
**≥30 ng/mL**	47 (37.0)	122 (48.6)	169 (27.8)	
**Two levels**				
**0–29 ng/mL**	80 (63.0)	129 (51.4)	209	0.04
**≥30 ng/mL**	47 (37.0)	122 (48.6)	169	
**Vitamin D ng/mL + SD ^1^**	29.0 ± 8.3	30.4 ±8.9		0.17

^1^ SD = standard deviation.

**Table 3 tropicalmed-10-00098-t003:** Vitamin D status and SARS-CoV-2 reinfections. Incidence rates. Cox proportional hazard models. Person-days. Incidence rate and 95% confidence interval (CI).

Vitamin D Levels	SARS-CoV-2 Reinfections	Person-Days	Incidence Rate 1000 Person-Days	95% CI
**<20 ng/mL**	11	22,074	0.50	0.28–0.90
**20–29 ng/mL**	69	138,627	0.50	0.39–0.63
**≥30 ng/mL**	47	126,578	0.37	0.28–0.49
**<30 ng/mL**	80	160,701	0.50	0.40–0.62
**≥30 ng/mL**	47	126,578	0.37	0.28–0.49
**Total**	127	287,348	0.44	0.37–0.53

**Table 4 tropicalmed-10-00098-t004:** Vitamin D status and SARS-CoV-2 reinfections. Cox proportional hazards models. Incidence rate, crude and adjusted hazard ratios (HRs), and 95% confidence interval (CI).

Vitamin D	Crude Hazard Ratios		Adjusted Hazard Ratios ^1^	
**Levels**	HR (95% CI)	*p*-value	HR (95% CI)	*p*-value
**<20 ng/mL**	1.25(0.65–2.42)	0.50	1.79 (0.89–3.59)	0.10
**20–29 ng/mL**	1.26 (0.87–1.83)	0.22	1.59 (1.06–2.38)	0.02
**≥30 ng/mL**	1.00		1.00	
**Trend**	1.17 (0.89–1.55)	0.26	1.42 (1.06–1.92)	0.02
**<30 ng/mL**	1.26 (0.88–1.82)	0.21	1.61 (1.09–2.39)	0.02
**≥30 ng/mL**	1.00		1.00	

^1^ Adjusted for age, sex, number of doses of SARS-CoV-2 vaccine, chronic disease, smoking, alcohol consumption, obesity BMI ≥ 30, family COVID-19 case, and high exposure to a COVID-19 case.

**Table 5 tropicalmed-10-00098-t005:** Vitamin D status and SARS-CoV-2 reinfections. Inverse probability weighting regression. Crude and adjusted reinfection incidence rates; 95% confidence interval (CI).

Vitamin D Levels	Crude Incidence Rate	95% CI	Adjusted ^1^ Incidence Rate	95% CI
**<20 ng/mL**	0.39	0.21–0.57	0.32	0.17–0.49
**20–29 ng/mL**	0.38	0.31–0.45	0.40	0.33–0.47
**≥30 ng/mL**	0.28	0.21–0.35	0.25	0.18–0.31
**Trend**	Z = 2.09	*p*-value = 0.04	Z = 2.54	*p*-value = 0.01
**<30 ng/mL**	0.38	0.31–0.45	0.40	0.33–0.47
**≥30 ng/mL**	0.28	0.21–0.35	0.25	0.18–0.32

^1^ Adjusted for age, sex, number of doses of SARS-CoV-2 vaccine, chronic disease, smoking, alcohol consumption, obesity BMI ≥ 30, family COVID-19 case, and high exposure to a COVID-19 case.

**Table 6 tropicalmed-10-00098-t006:** Vitamin D status and SARS-CoV-2 reinfections. Inverse probability weighting regression. Crude and adjusted relative risk (RR); 95% confidence interval (CI).

Vitamin D	Crude Relative Risk		Adjusted Relative Risk ^1^	
**Levels (ng/mL)**	RR (95% CI)	*p*-value	RR (95% CI)	*p*-value
**<20 ng/mL**	1.41 (0.84–2.38)	0.19	1.29 (0.74–2.22)	0.37
**20–29 ng/mL**	1.37 (1.01–1.87)	0.04	1.57 (1.15–2.16)	0.01
**≥30 ng/mL**	1.00		1.00	
**<30 ng/mL**	1.38 (1.02–1.85)	0.04	1.57 (1.15–2.15)	0.01
**≥30 ng/mL**	1.00		1.00	

^1^ Adjusted for age, sex, doses of SARS-CoV-2 vaccine, chronic disease, smoking, alcohol consumption, obesity, BMI ≥ 30, family COVID-19 case, and high exposure COVID-19 case.

## Data Availability

The data of the study can be consulted if the authors are requested.

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
