# Peer review of "Vitamin D Status and Incidence of SARS-CoV-2 Reinfections in the Borriana COVID-19 Cohort: A Population-Based Prospective Cohort Study"

_tropicalmed, 2025, doi:10.3390/tropicalmed10040098_

Round 1
Reviewer 1 Report
Comments and Suggestions for Authors
Relevance of the study. It is known, that in most cases the new coronavirus infection (COVID-19) caused by SARS-CoV-2 has a favorable prognosis. However, sometimes it can lead to severe systemic damage requiring hospitalization and even to death of the patient. Studies in recent years have shown a relationship between the level of serum 25-hydroxyvitamin D (25(OH)D) and the progression, severity and outcome of severe acute respiratory syndrome coronavirus-2 (SARS-CoV-2) infection. Previously published results from a number of meta-analyses indicate that individuals with vitamin D insufficiency, vitamin D deficiency, or severe vitamin D deficiency seem to be at higher risk of SARS-CoV-2 infection [Pereira M, Dantas Damascena A, Galvão Azevedo LM, de Almeida Oliveira T, da Mota Santana J. Vitamin D deficiency aggravates COVID-19: systematic review and meta-analysis. Crit Rev Food Sci Nutr. (2020) 4:1–9. 10.1080/10408398.2020.1841090 ; Chiodini I, Gatti D, Soranna D, Merlotti D, Mingiano C, Fassio A, Adami G, Falchetti A, Eller-Vainicher C, Rossini M, Persani L, Zambon A, Gennari L. Vitamin D Status and SARS-CoV-2 Infection and COVID-19 Clinical Outcomes. Front Public Health. 2021 Dec 22;9:736665. doi: 10.3389/fpubh.2021.736665. PMID: 35004568; PMCID: PMC8727532.; Liu N, Sun J, Wang X, Zhang T, Zhao M, Li H. Low vitamin D status is associated with coronavirus disease 2019 outcomes: a systematic review and meta-analysis. Int J Infect Dis. (2021) 104:58–64. 10.1016/j.ijid.2020.12.077]. Taking into account the high prevalence of vitamin D deficiency and insufficiency (the proportion of individuals with normal levels of provision does not exceed 20%), the question of the possible contribution of this nutrient to the prevention and treatment of COVID-19 is being actively studied. However, data on the impact of this factor on the frequency of recurrent infections are scarce, so this topic is of interest.
Purpose of investigation.
The purpose of this clinical trial was to investigate whether the risk of reinfection with SARS-CoV-2 is increased in individuals with suboptimal vitamin D status.
Methods. The internationally accepted method of determining the level of 25(OH)D (chemiluminescent immunoassay) and the proportional hazards model (Cox regression) were used to predict the risk of an event and to assess the impact of pre-specified independent variables (predictors) on this risk.
Results. The study included 378 subjects. In this cohort, 127 reinfections (33.6%) were recorded, the frequency of which depended on the 25(OH)D status (with a given confidence coefficient of 95%). The presented results of a prospective cohort study indicate that at a 25(OH)D level below the sufficient level (< 30 ng/ml), the risk of reinfection with SARS-CoV-2 increases. The authors make a reasonable conclusion, supported by the presented results, that correction and maintenance of the 25(OH)D level in the blood may have a preventive value and will reduce the frequency of reinfection with SARS-CoV-2.
Disadvantages.
The authors indicated the average age of the participants (38.8 ± 16.6 years). However, it fluctuated over a wide range from 1 to 75 years. From the data provided it is not clear how the frequency of reinfections was distributed in different age groups, and, accordingly, the basis for the conclusion that cases of reinfection were more often recorded for young people, is unclear, especially considering that vitamin D deficiency is diagnosed more often in the elderly.
The authors also noted that the frequency of reinfection did not depend on the presence of chronic diseases. However, the presence of specific chronic diseases was not indicated for the participants. This seems important, since it is well known that the presence of chronic diseases, such as diabetes and cardiovascular diseases, increases the risk of severity and mortality in COVID-19; gastrointestinal or kidney pathology can affect the values of 25(OH)D in the studied population. There is no information on the intake, doses and duration of vitamin D supplementation (if the level was corrected).
Note on text formatting: in the text of the article and in the list of references there are many places where there are no spaces (for example, lines 173, 393, 397, 401, 424,426, etc.)
Line 430 "...Justine, A. consensus....."- should be Consensus....
Author Response
Reviewer 1.
Thank you very much for the review and evaluation of our manuscript.
Relevance of the study. It is known, that in most cases the new coronavirus infection (COVID-19) caused by SARS-CoV-2 has a favorable prognosis. However, sometimes it can lead to severe systemic damage requiring hospitalization and even to death of the patient. Studies in recent years have shown a relationship between the level of serum 25-hydroxyvitamin D (25(OH)D) and the progression, severity and outcome of severe acute respiratory syndrome coronavirus-2 (SARS-CoV-2) infection. Previously published results from a number of meta-analyses indicate that individuals with vitamin D insufficiency, vitamin D deficiency, or severe vitamin D deficiency seem to be at higher risk of SARS-CoV-2 infection [Pereira M, Dantas Damascena A, Galvão Azevedo LM, de Almeida Oliveira T, da Mota Santana J. Vitamin D deficiency aggravates COVID-19: systematic review and meta-analysis. Crit Rev Food Sci Nutr. (2020) 4:1–9. 10.1080/10408398.2020.1841090 ; Chiodini I, Gatti D, Soranna D, Merlotti D, Mingiano C, Fassio A, Adami G, Falchetti A, Eller-Vainicher C, Rossini M, Persani L, Zambon A, Gennari L. Vitamin D Status and SARS-CoV-2 Infection and COVID-19 Clinical Outcomes. Front Public Health. 2021 Dec 22;9:736665. doi: 10.3389/fpubh.2021.736665. PMID: 35004568; PMCID: PMC8727532.; Liu N, Sun J, Wang X, Zhang T, Zhao M, Li H. Low vitamin D status is associated with coronavirus disease 2019 outcomes: a systematic review and meta-analysis. Int J Infect Dis. (2021) 104:58–64. 10.1016/j.ijid.2020.12.077]. Taking into account the high prevalence of vitamin D deficiency and insufficiency (the proportion of individuals with normal levels of provision does not exceed 20%), the question of the possible contribution of this nutrient to the prevention and treatment of COVID-19 is being actively studied. However, data on the impact of this factor on the frequency of recurrent infections are scarce, so this topic is of interest.
We appreciate your extensive comments on our manuscript. Certainly, there have been many studies on the vitamin D status and the COVID-19 outcomes, as the reviewer mentions. However, the incidence of SARS-CoV-2 reinfections considering the vitamin D status is less studied. On the other hand, there controversy about the role of vitamin D in SARS-CoV-2 infection, and we have included results pros and against our hypothesis.
Purpose of investigation.
The purpose of this clinical trial was to investigate whether the risk of reinfection with SARS-CoV-2 is increased in individuals with suboptimal vitamin D status.
The aim of our study was considering the effect of vitamin D status on SARS-CoV-2 reinfections, and the design of the study is a population-based prospective cohort.
Methods. The internationally accepted method of determining the level of 25(OH)D (chemiluminescent immunoassay) and the proportional hazards model (Cox regression) were used to predict the risk of an event and to assess the impact of pre-specified independent variables (predictors) on this risk.
We agree with the reviewer that multivariable Cox proportional hazard models can obtain a prediction of the risk of SARS-CoV-2 reinfection considering the vitamin D status.
Results. The study included 378 subjects. In this cohort, 127 reinfections (33.6%) were recorded, the frequency of which depended on the 25(OH)D status (with a given confidence coefficient of 95%). The presented results of a prospective cohort study indicate that at a 25(OH)D level below the sufficient level (< 30 ng/ml), the risk of reinfection with SARS-CoV-2 increases. The authors make a reasonable conclusion, supported by the presented results, that correction and maintenance of the 25(OH)D level in the blood may have a preventive value and will reduce the frequency of reinfection with SARS-CoV-2.
Thank you very much your evaluation of our results.
Disadvantages.
The authors indicated the average age of the participants (38.8 ± 16.6 years). However, it fluctuated over a wide range from 1 to 75 years. From the data provided it is not clear how the frequency of reinfections was distributed in different age groups, and, accordingly, the basis for the conclusion that cases of reinfection were more often recorded for young people, is unclear, especially considering that vitamin D deficiency is diagnosed more often in the elderly.
Thank you very much for your indication. In Table 1, we add the incidence of SARS-CoV-2 reinfection with age groups. The age group 1-24 years was the most affecte , followed by the other group. In general, medical publications (1-2) found that SARS-CoV-2 reinfections were more frequent in young adults, aged 18 and 40, followed by the group aged 41-60 years. These groups are highly socially active and have greater exposure to virus transmission with mild clinical episodes. This could suggest that vitamin D level may be insufficient to prevent reinfections. The group of 61 years old and more was less exposed to virus transmission, which may compensate for a greater deficiency in vitamin D in this age group.
The authors also noted that the frequency of reinfection did not depend on the presence of chronic diseases. However, the presence of specific chronic diseases was not indicated for the participants. This seems important, since it is well known that the presence of chronic diseases, such as diabetes and cardiovascular diseases, increases the risk of severity and mortality in COVID-19; gastrointestinal or kidney pathology can affect the values of 25(OH)D in the studied population. There is no information on the intake, doses and duration of vitamin D supplementation (if the level was corrected).
Thank you very much for the suggestion. Our manuscript reports the presence of chronic disease without specifying the type, and no differences in SARS-CoV-2 reinfection between the two groups of participants were found regarding chronic disease prevalence. In Table 1, we add a detailed analysis of several chronic diseases prevalence with respect to SARS-reinfections, and no significant differences were found. SARS-CoV-2 reinfections were more frequents in the young adults, while chronic diseases are more prevalent in the elderly, who had less virus exposure. In addition, the course of the reinfections was mild and less severe (3-4). In the other hand, our objective was to examine the effect of vitamin D status on SARS-CoV-2 reinfections, and the chronic diseases prevalence was adjusted for.
With respect to the information on the intake, doses and duration of vitamin D supplementation, it was considered that serum 25(OH)D levels determination was the best option to know the vitamin D status (5-6). There are wide variations in intake, doses, and duration of vitamin D, and adsorption, making it difficult to measure in relation to vitamin D status. In addition, the evaluation sunlight exposure or dietary intake of vitamin D is problematic (7).
Note on text formatting: in the text of the article and in the list of references there are many places where there are no spaces (for example, lines 173, 393, 397, 401, 424,426, etc.)
Line 430 "...Justine, A. consensus....."- should be Consensus....
Thank you very much for your indications. We have followed your indications in the text.
References
1.Nagao M, Matsumura Y, Yamamoto M, Shinohara K, Noguchi T, Yukawa S, Tsuchido Y, Teraishi H, Inoue H, Ikeda T. Incidence of and risk factors for suspected COVID-19 reinfection in Kyoto City: a population-based epidemiological study. Eur J Clin Microbiol Infect Dis. 2023;42:973-979.
2.Bastard J, Taisne B, Figoni J, Mailles A, Durand J, Fayad M, Josset L, Maisa A, van der Werf S, Parent du Châtelet I, Bernard-Stoecklin S. Impact of the Omicron variant on SARS-CoV-2 reinfections in France, March 2021 to February 2022. Euro Surveill. 2022 ;27:2200247.
3.Fabiánová K, Kynčl J, Vlčková I, Jiřincová H, Košťálová J, Liptáková M, Orlíková H, Šebestová H, Limberková R, Macková B, Malý M. COVID-19 reinfections. Epidemiol Mikrobiol Imunol. 2021;70:62-67.
4.Shaheen NA, Sambas R, Alenezi M, Alharbi NK, Aldibasi O, Bosaeed M. COVID-19 reinfection: A multicenter retrospective study in Saudi Arabia. Ann Thorac Med. 2022 ;17:81-86.
5.Vieth R. What is the optimal vitamin D status for health? Prog Biophys Mol Biol. 2006 ;92:26-32.
6.Grant WB, Boucher BJ, Bhattoa HP, Lahore H. Why vitamin D clinical trials should be based on 25-hydroxyvitamin D concentrations. J Steroid Biochem Mol Biol. 2018;177:266-269.
7.Ramasamy I. Vitamin D Metabolism and Guidelines for Vitamin D Supplementation. Clin Biochem Rev. 2020;41:103-126.
Reviewer 2 Report
Comments and Suggestions for Authors
poulation-based.
should be
population-based.
|
Levels of 25(OH)D were measured by electrochemiluminescence-based assay El- |
111 |
|
ecsys of Roche Diagnostic |
Comment: Please discuss its precision and accuracy.
Note that serum 25(OH)D concentration deceases after acute inflammatory illness. See this letter and articles that cited it.
Smolders, J.; van den Ouweland, J.; Geven, C.; Pickkers, P.; Kox, M. Letter to the Editor: Vitamin D deficiency in COVID-19: Mixing up cause and consequence. Metabolism 2021, 115, 154434
Did “after the first SARS-CoV-2 infection” mean after COVID-19 or merely infection without the disease?
If it meant infection with or without progressing to COVID-19, what are the findings regarding progressing to COVID-19? That would be more likely in the participants over the age of 60 years, of whom there appear to be very few, if any.
Serum VitD status was measured two times in October 2020 and June 2022.
All the participants had their VitD status measured 105 after the first SARS-CoV-2 infection.
|
For participants with 107 SARS-CoV-2 reinfections, the reported VitD status was the closest before the first reinfec- 108 tion and at least three weeks before this reinfection. The time between VitD status deter- 109 mination and SARS-CoV-2 reinfection or finishing the follow-up had a mean of 284±144.2 days. |
Comment: Most likely, seasonal adjustment of 25OHD concentrations should be done to make the study more accurate. See, e.g.
25-hydroxyvitamin D, IGF-1, and metabolic syndrome at 45 years of age: a cross-sectional study in the 1958 British Birth Cohort.
Hyppönen E, Boucher BJ, Berry DJ, Power C. Diabetes. 2008 Feb;57(2):298-305. doi: 10.2337/db07-1122.
How much did vaccine status reduce risk of reinfection? Table 1 indicates that the greater the number of vaccinations, the greater the probability of no reinfection. I think these data should be analyzed, not just reported.
Significant digits. The general rule is that no more non-zero digits should be given than are justified by the uncertainty of the value.
See "Too many digits: the presentation of numerical data"
https://www.ncbi.nlm.nih.gov/pmc/articles/PMC4483789/
If the uncertainty (or difference when comparing numbers) is greater than about 7%, only two non-zero digits are justified.
P values should be given to two decimal places unless the first two are 00 or the number lies between 0.045 and 0.054. If the first two are 00, then only one non-zero digit can be given.
Thus, p values should be adjusted
|
0.4983 |
0.2760-0.8998 |
Should be
|
0.50 |
0.28-0.90 |
|
<20 ng/ml |
0.393 |
0.212-0.573 |
Should be
|
<20 ng/ml |
0.39 |
0.21-0.57 |
|
Age (years) + SD1 |
37.9±17.0 |
Should be
|
Age (years) + SD1 |
38±17 |
Please review all numbers in abstract, text, tables, and figures and adjust accordingly.
Author Response
Review 2
Comments and Suggestions for Authors
Thank you very much for the review and evaluation of our manuscript.
poulation-based should be population-based.
Thank you very much your indication, and we have corrected the word.
|
Levels of 25(OH)D were measured by electrochemiluminescence-based assay El- |
111 |
|
ecsys of Roche Diagnostic |
|
Comment: Please discuss its precision and accuracy.
Thank you very much for your comment. In the discussion, we have included a limitation of our study, considering that the technique used for determining vitamin D level tends to underestimate the total 25-hydroxyvitamin D [25(OH)D] levels (1).In this context, the relationship between vitamin D status and SARS-CoV-2 reinfections could be decreased. The direction of this information bias would be towards the null hypothesis. Compared with the liquid chromatography tandem mass spectrometric (LC–MS/MS), the gold standard for the measuring of vitamin D status (2), the technique employed in this study has lower accuracy. However, it is considered acceptable for use in clinical laboratories, with an unlikely risk of misclassifying of vitamin D status (3).
Note that serum 25(OH)D concentration deceases after acute inflammatory illness. See this letter and articles that cited it.Smolders, J.; van den Ouweland, J.; Geven, C.; Pickkers, P.; Kox, M. Letter to the Editor: Vitamin D deficiency in COVID-19: Mixing up cause and consequence. Metabolism 2021, 115, 154434
We appreciate this indication. In fact, the 21 number of our reference list correspond to the reference mentioned by the reviewer. This is an important issue in many studies on the effect of vitamin D on infectious diseases.
Did “after the first SARS-CoV-2 infection” mean after COVID-19 or merely infection without the disease? If it meant infection with or without progressing to COVID-19, what are the findings regarding progressing to COVID-19? That would be more likely in the participants over the age of 60 years, of whom there appear to be very few, if any.
Thank you very much your question. The first SARS-CoV-2 infections were symptomatic in the majority of participants 89.9% (340/378), with 4.8% hospitalizations (18/378), and mild episodes. The course of the COVID-19 disease was towards recovery, although the incidence of sequelae was high 28.1% affected long COVID syndrome. The participants age group of 60 years and over was the most affected. We add these results in the manuscript.
Serum VitD status was measured two times in October 2020 and June 2022. All the participants had their VitD status measured 105 after the first SARS-CoV-2 infection.
|
For participants with 107 SARS-CoV-2 reinfections, the reported VitD status was the closest before the first reinfec- 108 tion and at least three weeks before this reinfection. The time between VitD status deter- 109 mination and SARS-CoV-2 reinfection or finishing the follow-up had a mean of 284±144.2 days. |
Comment: Most likely, seasonal adjustment of 25OHD concentrations should be done to make the study more accurate. See, e.g.25-hydroxyvitamin D, IGF-1, and metabolic syndrome at 45 years of age: a cross-sectional study in the 1958 British Birth Cohort.Hyppönen E, Boucher BJ, Berry DJ, Power C. Diabetes. 2008 Feb;57(2):298-305. doi: 10.2337/db07-1122.
Thank you very much for this interesting comment. The concentration of vitamin D is known to have seasonal variations. In our study, no significant difference of vitamin D concentration was observed between the first and the second determinations, suggesting that this variation does not effect of vitamin D status and the SARS-CoV-2 reinfections. In the other hand, the date of vitamin D measurement is not related per se with the SARS-CoV-2 reinfections, as these depend of virus circulation and exposure. In our Directed Acyclic Graph approach for controlling potential confounding factors, the dates vitamin D measurement were not associated with the risk of SARS-CoV-2 infections. Adjusting for this date would be considered unnecessary adjustment in controlling of confounding factors in the statistical models. It could introduce bias and imprecision (4-5).
How much did vaccine status reduce risk of reinfection? Table 1 indicates that the greater the number of vaccinations, the greater the probability of no reinfection. I think these data should be analyzed, not just reported.
Thank you very much for your suggestion. The effect of SARS-CoV-2 vaccines on COVID-19 disease and reinfections has been studied for different types of vaccines and SARS-CoV-2 variants (6-8). In addition, this effect could be modest when comparing COVID-19 recovered individuals and the COVID-19 naïve population (9). Our objective in this study was to measure the effect of vitamin D status on the risk of SARS-CoV-2 reinfections. The SARS-CoV-2 vaccines status was adjusted for in our multivariable models. Furthermore, no significant difference between vaccination status and SARS-CoV-2 reinfections was observed in the two groups. On the other hand, the study of vaccination and risk of reinfection needs different variables for the adjustment, because in our models the predictor variable is vitamin D status. We hope to investigate this interesting question in next studies.
Significant digits. The general rule is that no more non-zero digits should be given than are justified by the uncertainty of the value.See "Too many digits: the presentation of numerical data"https://www.ncbi.nlm.nih.gov/pmc/articles/PMC4483789/
If the uncertainty (or difference when comparing numbers) is greater than about 7%, only two non-zero digits are justified.
P values should be given to two decimal places unless the first two are 00 or the number lies between 0.045 and 0.054. If the first two are 00, then only one non-zero digit can be given.
Thus, p values should be adjusted
|
0.4983 |
0.2760-0.8998 |
Should be
|
0.50 |
0.28-0.90 |
|
|
|
<20 ng/ml |
0.393 |
0.212-0.573 |
|
Should be
|
<20 ng/ml |
0.39 |
0.21-0.57 |
|
|
Age (years) + SD1 |
37.9±17.0 |
|
|
Should be
|
Age (years) + SD1 |
38±17 |
Please review all numbers in abstract, text, tables, and figures and adjust accordingly.
Thank you very much for your comments. We are change the presentation of decimal numbers following your indications. However, we maintain the two decimal numbers for the confidence intervals and one decimal number for integers.
Thank you very much for your indication. We have change the values except per the p-values.
References
1.Altieri B, Cavalier E, Bhattoa HP, Pérez-López FR, López-Baena MT, Pérez-Roncero GR, Chedraui P, Annweiler C, Della Casa S, Zelzer S, Herrmann M, Faggiano A, Colao A, Holick MF. Vitamin D testing: advantages and limits of the current assays. Eur J Clin Nutr. 2020;74:231-247.
2.Binick S, Matthews SW, Kamp KJ, Heitkemper M. Vitamin D Measurement: Clinical Practice and Research Implications. J Nurse Pract. 2023;19:104481.
3.Geno KA, Tolan NV, Singh RJ, Nerenz RD. Improved Recognition of 25-Hydroxyvitamin D2 by 2 Automated Immunoassays. J Appl Lab Med. 2020;5:1287-1295.
4.Schisterman EF, Cole SR, Platt RW. Overadjustment bias and unnecessary adjustment in epidemiologic studies. Epidemiology. 2009;20:488-95.
5.Conroy S, Murray EJ. Let the question determine the methods: descriptive epidemiology done right. Br J Cancer. 2020 Oct;123(9):1351-1352.
6.Hammerman A, Sergienko R, Friger M, Beckenstein T, Peretz A, Netzer D, Yaron S, Arbel R. Effectiveness of the BNT162b2 Vaccine after Recovery from Covid-19. N Engl J Med. 2022;386:1221-1229.
7.Murugesan M, Mathews P, Paul H, Karthik R, Mammen JJ, Rupali P. Protective effect conferred by prior infection and vaccination on COVID-19 in a healthcare worker cohort in South India. PLoS One. 2022;17:e0268797.
8.Cegolon L, Larese Filon F. COVID-19 in City Council Civil Servants, 1 March 2020-31 January 2023: Risk of Infection, Reinfection, Vaccine Effectiveness and the Impact of Heterologous Triple Vaccination. Vaccines (Basel). 2024;12:254.
9.Shenai MB, Rahme R, Noorchashm H. Equivalency of Protection From Natural Immunity in COVID-19 Recovered Versus Fully Vaccinated Persons: A Systematic Review and Pooled Analysis. Cureus. 2021;13:e19102.
Round 2
Reviewer 2 Report
Comments and Suggestions for Authors
Please review all numbers in abstract, text, tables, and figures and adjust accordingly.
Thank you very much for your comments. We are change the presentation of decimal numbers following your indications. However, we maintain the two decimal numbers for the confidence intervals and one decimal number for integers.
Thank you very much for your indication. We have change the values except per the p-values.
Comment: Why were the p values not changed. They should be changed as stated.
Author Response
Review 2
Second revision
Please review all numbers in abstract, text, tables, and figures and adjust accordingly.
Comment: Why were the p values not changed. They should be changed as stated.
Thank you very much for your indication. We are reviewed all numbers in the manuscript, and we had included three decimals in the p-value because some journals accept them. Now, we have changed all the p-value to two decimals.